# Understanding the language barriers to translating informed consent documents for maternal health trials in Zambia: a qualitative study

Alice Beardmore-Gray  ,[1] Musonda Simwinga,[2] Bellington Vwalika,[3] Sebastian Chinkoyo,[4] Lucy Chappell,[1] Jane Sandall,[1] Andrew Shennan[1]

JS and AS are joint senior authors.

¹Department of Women and Children's Health, School of Life Course Sciences, King's College London, London, UK
²Zambart, Lusaka, Zambia
³University Teaching Hospital, Lusaka, Zambia
⁴Department of Obstetrics and Gynaecology, Ndola Teaching Hospital, Ndola, Zambia

**Correspondence to**
Dr Alice Beardmore-Gray;
alice.1.beardmore-gray@kcl.ac.uk

## ABSTRACT

**Objective** Providing comprehensible information is essential to the process of valid informed consent. Recruitment materials designed by sponsoring institutions in English-speaking, high-income countries are commonly translated for use in global health studies in other countries; however, key concepts are often missed, misunderstood or 'lost in translation'. The aim of this study was to explore the language barriers to informed consent, focusing on the challenges of translating recruitment materials for maternal health studies into Zambian languages.

**Design** We used a qualitative approach, which incorporated a multistakeholder workshop (11 participants), in-depth interviews with researchers and translators (8 participants) and two community-based focus groups with volunteers from community advisory boards (20 participants). Content analysis was used to identify terms commonly occurring in recruitment materials prior to the workshop. The framework analysis approach was used to analyse interview data, and a simple inductive thematic analysis approach was used to analyse focus group data.

**Setting** The study was based in Lusaka, Zambia.

**Results** The workshop highlighted difficulties in translating research terms and pregnancy-specific terms, as well as widespread concern that current templates are too long, use overly formal language and are designed with little input from local teams. Framework analysis of in-depth interviews identified barriers to participant understanding relating to design and development of recruitment materials, language, local context and communication styles. Focus group participants confirmed these findings and suggested potential solutions to ensure the language and content of recruitment materials can be better understood.

**Conclusion** Our findings demonstrate that the way in which recruitment materials are currently designed, translated and disseminated may not enable potential trial participants to fully understand the information provided. Instead of using overly complex institutional templates, recruitment materials should be created through an iterative and interactive process that provides truly comprehensible information in a format appropriate for its intended participants.

## STRENGTHS AND LIMITATIONS OF THIS STUDY

⇒ The use of a mix of qualitative data collection methods (interviews and focus groups) triangulated and enhanced the reliability of our findings and ensured representation of a broad range of perspectives.

⇒ Inclusion of additional stakeholders, such as members of ethical review boards, could have provided more information on the issue of informed consent, particularly regarding the ethical review process for recruitment materials.

⇒ The inclusion of community advisory board members strengthened our study by providing an important community voice; however, inclusion of a wider range of individuals from the community, including those more likely to be marginalised, and pregnant women in particular, could have ensured wider representation and added further to our findings.

⇒ The challenges described in this study are likely to be country and context specific. Our findings may inform other maternal health researchers working in Zambia, as well as outlining important principles which may apply to similar settings.

## INTRODUCTION
### Statement of the problem

Global health research typically involves partnerships between high-income and low or middle-income countries. These partnerships can sometimes perpetuate inherent structural inequalities or power dynamics,[1–5] whereby research methodology and institutional processes designed in a high-income country may be imposed on low-income partners without considering the relevance or acceptability to the local population. The process of informed consent, and ethical review of consent documents, are two of the domains which may be affected by this imbalance. This study evaluates an example within the context of a maternal health trial conducted in Lusaka, Zambia, specifically exploring how language barriers, and issues surrounding

translation of recruitment materials, may impact on informed consent.

Informed consent is fundamental to any research involving human beings. For consent to be valid, participants must have the capacity to consent, act voluntarily and be provided with sufficient comprehensible information. These principles are well described and upheld by international ethical and legal frameworks.[6 7] However, these frameworks are based on knowledge systems generated and perpetuated by dominant groups in high-income countries[1] and often imposed on other communities without considering local expertise. The participant information leaflets and consent forms (recruitment materials) used for enrolling participants into clinical trials conducted in low or middle-income countries are often designed by sponsoring institutions based in high-income countries[8] and are therefore more likely to meet the needs of trial sponsors and ethical review boards, rather than those of the intended participants. There is a focus on written documentation, complex medicolegal language and lengthy forms providing excessive information. These forms are then translated via a process of forward and back translation into the local language(s) of the country where the research is taking place. However, a 2014 review into participant comprehension found that the majority of trial participants across different African countries did not understand several key domains of informed consent such as voluntariness, confidentiality and the difference between taking part in research and seeking medical care.[9] This is attributed to a lack of conceptual equivalence,[10–13] arising from a lack of directly equivalent terms, as well as languages that are predominantly spoken and therefore do not have standardised written formats. Use of overly complex words and medical terminology further exacerbates this lack of understanding.[14] Studies have also highlighted a lack of universal tools for assessing understanding of trial participants[9 15–17] and this in itself presents a barrier to identifying areas for improvement. Several studies have highlighted linguistic factors as a significant barrier to comprehension, but there is very little literature exploring this particular issue. Maternal health is a key research priority which justifiably attracts large numbers of research studies. However, pregnant women are a vulnerable population and in many low or middle-income countries, including Zambia, vulnerability may be compounded by low levels of educational attainment and literacy.[18] By exploring the language barriers to cross-cultural adaptation of recruitment materials for a maternal health-related clinical trial, we aim to improve the quality of recruitment materials provided to future participants in maternal health studies in Zambia, and to contribute towards local efforts to strengthen research ethics capacity, which has been identified as a key priority by the Zambia National Health Research Policy and the Zambian National Health Regulatory Authority.[19]

## Research objective

The overall aim of this study is to understand the language barriers to informed consent, and to demonstrate, via the example of translating maternal health research materials in Zambia, the importance of developing informed consent processes and providing participant information in a way that truly suits the needs of research participants.

## METHODS

We used a qualitative study design incorporating a participatory workshop, in-depth interviews and focus group discussions. This study took place in three phases (table 1), based primarily in Lusaka, Zambia, alongside a timing of delivery in pre-eclampsia trial[20]; the research was led by the coordinator of this trial, a UK doctor. A Standards for Reporting Qualitative Research checklist is provided in the online supplemental materials.

### Sampling strategy and data collection methods

The cross-sectional sample of recruitment materials used during phase 1 was obtained by inviting researchers working in Zambia to submit English language examples of recruitment materials they had previously developed (and subsequently translated) to inform individuals considering participation in their research studies (predominantly clinical trials). Researchers were identified via ongoing research being conducted at University Teaching Hospital, Lusaka, ongoing research conducted by the Department of Women and Children's Health at King's College London and via the Global Women's Research Society international conference. Researchers were asked to provide English language versions of participant information leaflets which were collated, read and analysed by the study lead (AB-G). All relevant examples of recruitment materials provided were included in the total sample of 13 documents. Summative content analysis (see the Data analysis section) was used to identify the most commonly occurring terms related to research and pregnancy (details shown in online supplemental table 1). These terms were organised into relevant themes such as pregnancy-specific terms, research concepts and confidentiality (online supplemental table 1). The workshop focused on how these different English terms could be translated for a Zambian population, and the potential difficulties that might be encountered when doing so. Through our discussion with workshop participants, we were able to identify which commonly occurring terms were most difficult to translate. Contemporaneous group notes were made on flip charts during this process. In addition, the independently performed back translations of participant information leaflets (translated as part of the timing of delivery in pre-eclampsia trial conducted in Zambia) were also discussed. These leaflets had been translated from English into Nyanja, and then back to English. The discussion focused on comparing and contrasting the original English versions with the back-translated versions.

**Table 1** Study phases and participants

| Phase | Activity | Participant summary |
|---|---|---|
| Phase 1 Lusaka 18 November 2019 | Facilitated workshop with invited participants from a variety of professional backgrounds. We set out to explore how key maternal health research terms, identified from a cross-sectional sample of recruitment materials, from different research studies, might be translated from English into Nyanja and Bemba and how this process might alter their meaning, as part of an initial exploratory exercise to guide the subsequent two phases. | There were 11 participants including AB-G (study lead). Five participants were female, and six were male. Nine were Zambian, two were British. Four were obstetric researchers, three were research assistants and four were translators with a background in teaching and social science. Participants were invited based on their ongoing involvement with a clinical trial evaluating timing of delivery in pre-eclampsia. |
| Phase 2 13 May to 1 July 2021 | In-depth interviews with key informants to understand in more detail the challenges involved with translating consent documents for a Zambian population. | A total of eight interviews took place. The age range of participants was 30–69, three were female and five were male. Most (six) had degrees, two had diplomas. Their occupations included language teacher (three participants), research coordinator (four participants) and one community engagement officer. |
| Phase 3 21 and 29 June 2021 | Focus group discussions with local community advisory board members at primary health clinics to interrogate findings from phases 1 and 2 with individuals who would be representative of potential research participants. | A total of two focus group discussions (20 participants in total, 10 in each group) took place. The mean age of participants was 28 years, 12 were female and 6 were male (information not provided for two participants). Eight participants had attended tertiary-level education, with the remainder having attended secondary-level education. |

Phase 2 in-depth interviews with key informants were significantly delayed due to the COVID-19 pandemic. An initial convenience sample of key informants was used, whereby individuals were invited to participate if they had prior experience of either translating recruitment materials or enrolling participants into research studies. As data collection continued, additional informants were invited to participate via snowball sampling. This comprised inviting other individuals, suggested by key informants, who were likely to have relevant insight and expertise, such as community engagement officers or research assistants. Interviews were conducted in English, the working language in Zambia, by the study lead (AB-G). A semi-structured interview guide was used (online supplemental table 2) and each interview was audio recorded and then transcribed. The interviews took place at times and locations convenient to the participants, primarily office spaces and meeting venues in Lusaka, Zambia. Phase 3 focus group discussions with community advisory board members were facilitated by three of the professional language teachers/translators who had participated in the in-depth interviews (phase 2) as key informants, supported by the study lead (AB-G). Focus group participants were invited by asking community advisory board members to participate if they wished. Community advisory board members were volunteers from the local community who were part of pre-existing community groups, linked to primary healthcare facilities in Lusaka, Zambia. Community advisory board members linked to Kanyama first-level hospital and Chawama first-level hospital were selected as these are two of the busiest primary healthcare facilities in Lusaka and both facilities had enrolled participants into the previously mentioned timing of delivery in pre-eclampsia trial. Invitations were sent out to community advisory board members via text message, and responding individuals were then invited to participate in a focus group discussion. Two initial focus groups were planned, as a purposeful sample, designed to interrogate findings from the workshop and key informant interviews. A focus group guide was developed following the phase 2 interviews and adapted from the interview topic guide (online supplemental table 2). Focus group discussions took place in outdoor meeting spaces attached to two first-level hospitals (Kanyama and Chawama) in Lusaka, and were audio recorded and transcribed. Focus groups were conducted in a mixture of English, Nyanja and Bemba and were translated at the time of transcription by a Zambian research assistant.

## Ethical considerations

Written informed consent was sought from all participants before any interviews or focus group discussions were conducted and participation in the study was entirely voluntary. Electronic copies of interview and focus group transcripts were stored on a password-protected hard drive. Participants were anonymised and referred to by initials or numbers only.

## Data analysis

Content analysis was used to analyse recruitment materials as a recognised method of rapidly identifying commonly occurring language. A summative approach was taken,

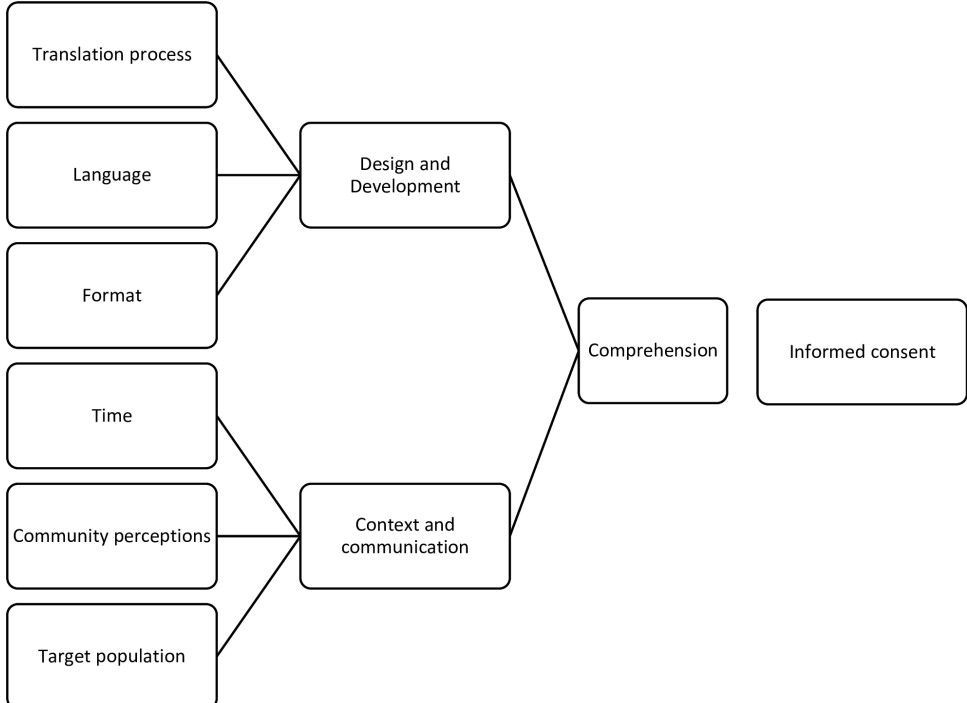

**Figure 1** Thematic framework.

identifying the number of times that pregnancy-related phrases and research terms arose in the sample of recruitment materials.[21] During the phase 1 workshop (conducted in English), participants discussed the interpretation of these commonly occurring English terms, and explored how they might be translated into Nyanja and Bemba. Key outputs from the discussion notes were used to inform the design of phases 2 and 3. Interview transcripts from phase 2 were uploaded to NVivo V.12 for data coding. A framework analysis approach was used to analyse interview data.[22] The theory underpinning this framework was drawn from the conceptual framework for the process of obtaining informed consent outlined by Bhutta,[8] theories of reading and language proficiency[23] and models for translation and cross-cultural adaptation such as those outlined by Brislin[24] and Flaherty *et al*.[25] These theories were combined into one overarching framework (online supplemental figure 1[14 18 24 25]) which guided data analysis, and was developed further during the analysis process, informing the final thematic framework shown in figure 1. Focus group data were analysed using a simple inductive thematic analysis approach. The themes identified were compared and contrasted to findings from the interview data. By collecting data using different methods (workshop, interviews and focus group discussions) and from different sources (eg, research professionals and community members) we were able to triangulate our data[26] and test the validity of our findings from each phase, thereby enhancing the trustworthiness of our data. Focus group discussions with community members were chosen as a method of interrogating the findings from the workshop and interviews, and to seek differing perspectives and suggestions from individuals

likely to represent potential research participants (as part of a local community linked to primary healthcare facilities involved in recruiting to clinical trials).

### Role of the funding source
The funder of the study had no role in study design, data collection, data analysis, data interpretation or writing of the report. The corresponding author had full access to all the data in the study and had final responsibility for the decision to submit for publication.

### Patient and public involvement
Patients and the public were not involved in the design or implementation of this study. However, the initial phase, a multidisciplinary workshop, incorporated individuals from a range of professional backgrounds including social scientists and language teachers in addition to clinicians and researchers. Subsequent phases involved professional translators and members of the public in the form of community advisory boards, with the overall findings and recommended actions reflecting their lived experience and perspectives.

### RESULTS
#### Phase 1: initial workshop
During the phase 1 workshop, participating translators highlighted the lack of equivalent terms (in Nyanja or Bemba) for words such as 'pre-eclampsia', 'proteinuria' and 'contractions', as well as differing interpretations of words such as 'research', 'benefits' and 'risks'. The word 'consent' itself was also raised as a term which could be interpreted differently depending on the context, with

**Table 2** Examples of back translations

| Original (English) | Back translation (from Nyanja translation) |
| --- | --- |
| We appreciate your time and are grateful for your help. However, there will not be any financial compensation for taking part in this study. By choosing to take part in this study you will be helping us to help other women like you in the future. | We are very thankful for giving us your time and all the help that you have rendered to us. Even if things are like this, there will be no funerals of any kind because you have taken part in this research study. |
| If you take part in the study we will collect some personal information. This will only be used by members of the research team if they need to contact you. This information will be kept confidential. This means that only members of the research team will have access to it, and it will not be shared with anybody else. The data will be protected according to UK Data Protection Laws. | I agree that my suggestions that I will give should be made use of in this lesson (the way my suggestions have been presented). I am aware that my suggestions will be kept following the best recommended practices of keeping secrets. |
| The CRADLE-4 Trial (Phase 1): the feasibility and acceptability of planned early delivery in pre-eclampsia in a low and middle-income setting. | Reviewing the advantages of early childhood delivery on the poor women and those at the centre of fending for their families. This is usually centred on the women with complications of swelling of feet and other body parts, excess proteins in the blood and urine and high blood pressure. |
| You have been invited to take part in this study because you have pre-eclampsia, but your condition does not require that your baby be delivered immediately. | Therefore, you are requested to take part in this study so as to help us get the facts regarding this matter. |

some communities being less familiar with the concept of individualised consent than others. Important examples of discrepancies identified during discussion of the back translations are shown in table 2. This interactive workshop highlighted several concerns regarding current procedures for designing and translating research documents, and the lived experiences of the participants suggested this was a common and widespread issue. The group therefore proposed further exploration of the language barriers to adaptation of maternal health recruitment materials in Zambia via in-depth interviews, followed by community focus group discussions to develop locally driven solutions that may be generalisable to other researchers working in similar settings.

### Phases 2 and 3: in-depth interviews and focus group discussions

The initial theoretical framework was modified throughout the coding process, with the final thematic framework used for data analysis shown in figure 1.

### Design and development of recruitment materials

The interview participants working as research coordinators felt they were not given sufficient opportunity to contribute to the design of recruitment materials at an early stage, stating that they are often invited to review documents only after they have already been finalised and submitted to the ethics committee (table 3, quote D1). Information leaflets were criticised as being too long and wordy, with emphasis placed on the need to present key messages more succinctly using alternative methods such as flyers, community announcements and household visits. The translation process itself was identified as a significant issue, due to an overemphasis on literal word for word translations, rather than communicating

the true meaning of the information. This issue was felt to be exacerbated by poor interactions between researchers and translators. The professional translators interviewed spoke of pressure to produce work within a tight timeframe, compounded by a lack of face-to-face meetings with researchers, meaning that research principles and scientific concepts were often not thoroughly understood by the individual translating the document (table 3, quote D2). Language itself was an important barrier, primarily due to a lack of equivalence—often there is simply no equivalent word in the local language for a particular English medical term. As a result, translators may try to explain the term using multiple words and phrases which ultimately distort or change the meaning (table 3, quote D3). Furthermore, a clear distinction was made between 'play' language and 'formal' language with some translators criticising the overuse of formal language in translated documents, rendering them incomprehensible to the intended recipients who use different, more colloquial versions. Finally, the presence of multiple languages in Zambia (72 in total) was identified as a further challenge, as most documents will be translated into just a few of these languages which will be understood to varying degrees by different individuals depending on their family background and where they live.

Focus group participants also felt that information in recruitment documents should be shortened and simplified and that lengthy information relating to the sponsoring institution and data protection was not necessary. Many participants felt that verbal explanations, audiovisual aids and flip charts could enhance information provision but agreed with interview participants that written documentation was an important component of the process that should not be eliminated. Participants

**Table 3** Illustrative quotes

| Design and development | |
|---|---|
| D1. 'What I noticed is that we just receive the consent, you can't change anything in the consent.' | Interview participant |
| D2. 'There are people, some people, would have sent work, you work on their consignment, you just send back. You've never met face to face. They have no time to sit with you.' | Interview participant |
| D3. 'Pre-eclampsia in our local language, we don't have it, it's not there, so a translator needs to have a rich vocabulary and full understanding for you to come up with the correct translation.' | Interview participant |
| D4. 'Here in Lusaka they don't use kubeleka, but instead they say (abala) so for this word, it will be difficult for the community to understand.' | Focus group participant |
| **Context and communication** | |
| C1. 'You will find that some people, when they find these women who maybe can't read on their own and they have to read for them, so you will find that most of the time, there is this issue of inadequate information being given and it will be like fast done.' | Interview participant |
| C2. 'You need to get consent from the husband and yet the pregnant woman is an adult, so they can consent on their own but they will not consent, they want consent from their husband or from their parents.' | Interview participant |
| C3. 'People need to understand, what is ultrasound, what is this machine, why are you doing this on me? What is its effect.' | Interview participant |
| C4. 'HIV, where you are doing blood draws so they would, from communities, they would, they would think you are selling their blood.' | Interview participant |
| C5. 'Looking at the community where we come from, the people that read this information trust me, most of them can't read, most of them can't even read the local language.' | Focus group participant |

felt that greater emphasis needed to be placed on the voluntary nature of study participation, with statements such as 'you do not have to take part if you do not want to' given greater prominence and translated clearly and directly to ensure the meaning was clear. Participants also stated that language related to funding needed to be clarified, as direct translation of the English phrasing implied possible financial incentives could be provided by taking part. Participants also provided examples of different terms that may be used to explain pregnancy or birth depending on the context, and that while informal terms were sometimes considered less 'respectful', they were often better understood by their community (table 3, quote D4). There was tension between some translators who wished to preserve the formal, grammatically correct version of their language as taught in schools and focus group participants who preferred more colloquial terms. A suggested solution was using more informal terms in brackets so that both the official and colloquial terms could be presented and communicated effectively, depending on the user. Focus group participants also suggested creating a glossary of certain words at the start of any document, using local terms to explain in detail medical terms such as pre-eclampsia or proteinuria for the reader. Participants expressed specific preferences for different translations of particular words, examples of which are presented in online supplemental table 3. Throughout, more informal versions were preferred, and alternative terms suggested which were sometimes different from the versions originally provided by translators.

### Context and communication

The way in which information is communicated to participants, as well as the context into which it is being delivered, was highlighted by both interview and focus group participants as an important area needing improvement. Some interview participants felt that potential participants are not given sufficient time to consider the information provided, with decisions often expected on the same day that a study is explained for the first time by research teams. Furthermore, some researchers described often needing to verbally explain recruitment materials to illiterate participants. They felt that this makes it difficult to standardise the information provided to potential participants and risks potential participants receiving insufficient or even inaccurate information (table 3, quote C1). When considering the context into which translated documents are being introduced, all of the interview participants raised the importance of the target population and the need to consider the levels of literacy, the languages used, the age and gender of potential participants (eg, many pregnant women require their husband's consent before participating in any study) and also the common misconceptions that may be prevalent within that community surrounding healthcare interventions or research studies (table 3, quotes C2–C4). Many interview participants highlighted the fact that use of inappropriate language or poorly designed forms will compound this issue, and risks both limiting the number of potential participants enrolled into a study and undermining the validity of the informed consent of those who do decide to take part.

Focus group participants raised similar concerns, recalling having previously been given brochures or leaflets to read, and not having the time or inclination to do so. Having more in-depth discussions, with audiovisual aids, and the opportunity for further discussions to ask questions at a later date were suggested as measures that may improve participant comprehension. Consistent with interview findings, focus group participants highlighted the importance of understanding the target community and in particular mentioned the fact that, in their experience, most individuals in their community could not read the local language (table 3, quote C5). They felt that simple information should be provided in ways that are easy to understand such as flip charts and pictures. However, the background and education of potential participants was also highlighted as an important factor to consider when choosing the most appropriate information format—with participants suggesting that in some communities, video consent may be deemed suspicious or inappropriate. Geographical region was also highlighted as important, with preferred terms changing depending on which area of the country the research is being conducted.

## DISCUSSION

Our collaborative workshop highlighted the discrepancies between the original English versions of recruitment materials and translated copies, as well as the difficulty in finding equivalent terms to accurately convey the intended meaning of key research concepts and medical words such as 'pre-eclampsia'. We identified several barriers to participant comprehension and informed consent within in-depth interview data, including a lack of time available to translators, poor literacy and rushed interactions between researchers and potential participants. Researchers working in Zambia felt that the content and layout of recruitment materials were designed by 'the owners' in English-speaking countries and that they had little opportunity to influence the design or make their voices heard, with translations subsequently regarded as poor quality. In contrast to the grammatically correct, formal translations often used by professional translators, focus group participants expressed a clear preference for translated versions of recruitment materials to use more informal language, and that this should vary depending on the target population of a study. Furthermore, while workshop participants suggested audiovisual aids as a potential solution, interview and focus group participants felt that although they may be a helpful supplement, it was important to have hard copies of written information to refer back to and maintain trust.

Previous research on informed consent has focused primarily on identifying gaps in participants' understanding and evaluating community perceptions of research. Our findings correlate with those described by other studies, which found that there were widespread misconceptions regarding the purpose of research, the benefits and risks of taking part and the use of research samples such as blood samples.[27 28] If the content of research documents does not address peoples' fears and beliefs (for example around blood tests or ultrasound scans) and explain in detail what is expected of participants and why, participants may base their decision on whether to participate or not on misinformation. Previous studies highlighted a need to further investigate the language barriers to effective communication about research, as well as to develop pretested and standardised tools that can be used to explain research concepts in a way the local community can understand. However, ours is the first study, to our knowledge, which explores these barriers, with a focus on translating recruitment materials and a specific focus on maternal health terms. We therefore build on the issues raised by previous work, exploring the specific difficulties relating to language and conceptual equivalence in more detail, adding voices from a cross section of individuals in Zambia, directly involved in the design and implementation of maternal health research, as well as community representatives of target populations.

There has been a call to action within the global health community to redress the systemic imbalances that are perpetuated by Eurocentric institutions and practices.[29] However, there are very few worked examples that demonstrate how these inequities may cause harm to research participants, and even fewer examples that suggest ways of dismantling these practices.[30] This study provides a practical and tangible example of ways in which researchers and ethical review boards can begin the process of change right away. A recent scoping review highlighted the financial, administrative and regulatory barriers to good quality ethical review in low and middle-income countries[31]; our study provides relevant findings that may be used to address some of these concerns. A collaborative, multidisciplinary research programme in Kenya has successfully implemented a systematic approach to translating contextualised informed consent templates, drawing on community engagement processes within their research programme, which has received positive engagement from researchers and ethics committees.[32] We present our own summary of our recommended actions for institutions, researchers and translators in figure 2, which represents the perspectives of the Zambian participants in this study, and could be used to inform a similar approach in a Zambian setting.

### Strengths and limitations

The initial research question and subsequent study design were influenced by the experiences of the study coordinator (AB-G) when translating recruitment materials for the feasibility study informing the main timing of delivery in pre-eclampsia trial,[20 33] which suggested the specific difficulties encountered during this process may represent a wider issue. This was explored further during the course of data collection and analysis, acknowledging the potential biases that may have been carried

| Research institutions and ethics committees | Researchers | Translators |
|---|---|---|
| • Adopt a more flexible and adaptive approach to templates<br>• Suport research teams to develop recruitment materials that are context-specific<br>• Ensure study protocols allow sufficient time and funding to support a robust translation process and consent process including community engagement activities<br>• Ensure strong oversight mechanisms to verify the quality and appropriateness of translated materials<br>• Support further research into alternative methods of providing participant information, such as pictures and videos | • Set aside sufficient time and funding to develop recruitment materials<br>• Meet face to face with translators and local language experts, ensuring the true meaning of recruitment materials can be understood<br>• Involve community representatives and local researchers from the outset, piloting early versions of translated materials and responding dynamically to feedback<br>• Move away from lengthy word documents with information that may be considered irrelevant by potential participants<br>• Consider a glossary of key terms at the start of any document, using simple and informal terms to explain important concepts or medical terms<br>• Consider the most appropriate format for the intended recipients, including flip charts and videos if appropriate | • Move away from literal, word for word translations<br>• Explore, and be guided by, local dialects and preferences for more informal language |

**Figure 2** Summary of recommended actions.

forward from this initial experience. Collecting data from different sources helped counteract any inherent individual bias. For example, the assumption that participants might prefer information provided in alternative formats was dispelled by both interview and focus group participants who felt it was important to have a written, hard copy of any recruitment materials. In their position as a trial coordinator, it is possible that interview and focus group participants may have viewed them as possessing a certain level of authority, and this in turn may have influenced the responses of the participants. Steps taken to counteract this included a relaxed communication style during interviews and using local translators to help facilitate focus group discussions. In their position as a researcher based in Zambia during the time period that this study took place, the study lead was able to connect with and seek out key informants within the local research community and seek guidance from local experts working in social science research. Language teachers and translators represented an important group of participants for this research. While they had previous professional experiences of translating research materials, it was clear that the objective of this study was to learn from and understand their experiences, rather than engage them in a professional capacity, thus limiting the potential for any

conflict of interest. The translators who performed the initial translations used for the back translations discussed at the workshop each worked as professional teachers of either Nyanja, Bemba or both, in the public education system in Zambia. Each translator had at least 5 years of experience of translating research documents for clinical trials. Although some of the discrepancies identified may be related to errors, rather than specific language barriers (for example, the addition of the word "funerals" in the first example provided in table 2), this highlights the importance of performing back translation (not always required by ethical review bodies), and allowing sufficient time for translators and researchers to meet face to face and discuss their work, an important process which, according to the translators interviewed, was frequently ignored by researchers. Focus group participants were recruited from local community advisory boards. These groups are local volunteers who are often consulted to gain community input and perspectives on healthcare interventions and research studies. While this meant they were well placed to participate in the focus group discussions facilitated as part of this study, participants outside of this well-established model may have provided a wider array of insights.

The views of both interview and focus group participants likely represent an urban population, though many interview participants had experience of a wide range of research studies conducted over different time periods and in different areas of the country. Interview and focus group participants had many experiences of research, given that they lived in Lusaka, the capital city, where many of the healthcare facilities have ongoing involvement in several research projects. A more remote setting in areas where participants are less familiar with research may have provided different findings. However, given the aim of this study was to specifically explore issues when translating, using and understanding participant information documents, the selected population was likely appropriate for the research objectives. Inclusion of ethical review board members, or study principal investigators, who are responsible for approving many of the recruitment materials used in global health studies, could have added an additional and important perspective on the issues explored in our study. Engaging these key stakeholders would be important in any future research and when implementing our recommended actions.

In this study, we focus on the example of translating recruitment materials for a maternal health-related clinical trial in Zambia. It is possible that the specific challenges described by participants in this study may not necessarily apply to other study designs or contexts, thereby limiting the generalisability of our findings. However, by sharing the key learning points identified from this qualitative study, it may prompt any individual involved in translating or using participant information in similar settings to critically review the language they use, and whether it is appropriate and comprehensible to its intended audience.

## CONCLUSIONS

Our study has identified that current methods of designing and translating recruitment materials for potential research participants in maternal health studies in Zambia may not always facilitate true understanding, and therefore may not serve the needs of their intended recipients. This problem requires researchers and ethics committees to re-evaluate their current practice and move away from viewing translation as merely a tick box exercise required to gain ethical approval, but a collaborative and dynamic process that can be adapted to suit the needs of the communities, countries and languages in which the research is taking place.

**Acknowledgements** We thank all the individuals who participated in this study, providing their valuable time and insight.

**Contributors** AB-G, AS and LC were involved in the study conception and in securing funding for the study. AB-G, MS, BV, SC, LC, JS and AS designed the study protocol and secured ethical approval for the study. AB-G coordinated the study supported by BV and SC. AB-G did the study analysis with input from MS and JS. AB-G wrote the original manuscript draft. All authors reviewed, contributed to and approved the final version of the manuscript. AB-G (guarantor) accepts full responsibility for the work and the conduct of the study, had full access to the data, and controlled the decision to publish.

**Funding** This study was funded by the King's ODA Research Partnership Seed Fund (KODA_1819_002). JS is a National Institute for Health Research (NIHR) senior investigator and is supported by the NIHR Applied Research Collaboration South London (NIHR ARC South London) at King's College Hospital NHS Foundation Trust.

**Disclaimer** The funder of the study had no role in study design, data collection, data analysis, data interpretation or writing of the report. The views expressed are those of the author(s) and not necessarily those of the NIHR or the Department of Health and Social Care.

**Competing interests** None declared.

**Patient and public involvement** Patients and/or the public were not involved in the design, or conduct, or reporting, or dissemination plans of this research.

**Patient consent for publication** Not applicable.

**Ethics approval** This study involves human participants and was approved by King's College London (MRSP-20/21-22350) and University of Zambia Biomedical Research Ethics Committee (1517-2020). Participants gave informed consent to participate in the study before taking part.

**Provenance and peer review** Not commissioned; externally peer reviewed.

**Data availability statement** Data are available upon reasonable request. The dataset will be available to appropriate academic parties on request from the corresponding author in accordance with the data sharing policies of King's College London, with input from the coauthor group where applicable.

**ORCID iD**
Alice Beardmore-Gray http://orcid.org/0000-0001-9923-4912

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
