## [Reviewer comments · BMJ Open]

ARTICLE DETAILS

TITLE (PROVISIONAL)	Understanding the language barriers to translating informed consent documents for maternal health trials in Zambia: a qualitative study
AUTHORS	Beardmore-Gray , Alice; Simwinga, Musonda; Vwalika, Bellington; Chinkoyo, Sebastian; Chappell, Dr Lucy; Sandall, Jane; Shennan, Andrew

VERSION 1 – REVIEW

REVIEWER	Pulford, Justin Liverpool School of Tropical Medicine, Department of International Public Health
REVIEW RETURNED	05-Sep-2023

GENERAL COMMENTS	Title The title is too broad as stated as it implies an extensive evaluation of the barriers to informed consent, when the focus is specifically on the issue of translating informed consent documents (primarily) within the context of maternal health research trials. Abstract Based on points made below in response to the substantive text, some additional context may need to be included in the abstract (e.g. the explicit focus on trials) and some statements toned down. The authors should also include the number of participants and data analysis approaches in the 'methods' sub-section. Introduction It is still not entirely clear whether the authors are focusing on translation of informed consent materials in a trial context or in global health research more broadly. Much of the argument is strongest within a trial context, e.g. the use of 'medico-legal language' is perhaps most pronounced in medical trials as compared to other study designs/research foci, and the authors specifically refer to 'trial sponsors' and 'trial participants' in places. However, there is also reference to a 'maternal health' focus (which includes, but is not limited to, research trials). If the work primarily relates to medical trials conducted within a maternal health context, then this needs to be made explicit from the outset. I would suggest making this statement within the sentence beginning 'This study evaluates...' (line 59) and possibly within the manuscript title. Having made this focus explicit, the authors can then frame their argument within this context and, where appropriate, note issues and findings that may be relevant to informed consent processes more broadly (rather than imply stated issues are relevant in all cases of informed consent
---

irrespective of research design/study focus). This would prevent/reduce some of the over generalisation that is still present when the explicit context is not made clear (i.e. informed consent language and protocols are often more strict within a medical trial context and issues associated with translating complex medical terminology are often more apparent in these cases). Other examples of over generalisation (not related to study design) are also apparent. For example, not all consent forms need to be translated (depends on country, type of participants etc).

Methods

The focus of the paper is specifically on translating informed consent documents within a Zambian context; however, in the 'sampling strategy' section it is stated that 'recruitment materials' were obtained from researchers working in Zambia and 'neighbouring countries'. Why were documents from neighbouring countries included and would the workshop participants have been able to critique these, especially any related back translations?

The data collection methods/procedures for the stage one workshop are not well described in the methodology. Rather, the description extends only to the sampling strategy with details re workshop process coming later in the 'results' section (e.g. this is where we are first introduced to the review of back translated documents).

In the 'data analysis' section the authors state that 'no formal analysis of workshop data was performed', yet (in addition to findings from the content analysis) a summary of key points from the workshop discussion and examples of flawed back translations emanating from the workshop are presented in the results. As this is presented as data then how it was derived needs to be made clearer in the methods (see previous point). I.e. what you 'did' in the workshop needs to be described in more detail in the methods and text pertaining to the workshop in the results section should be limited to what you 'found'.

Results

The content under 'phase one: initial workshop' should be amended based on comments above (i.e. content describing process moved to the methods section). Content in remaining sections OK as is.

Discussion

A lot of the content under 'strengths and weaknesses' is related to the issue of positionality in a qualitative research context. Positionality statements would typically be found in a the 'methods' section. Thus, the authors could consider shifting some of the material from here to a specific 'positionality' sub-section in the methods.

The issue of context could also be discussed in this section. I.e. if this study is primarily framed within the context of informed consent carried out in maternal health trials, then to what extent might the findings relate to informed consent in other study designs/study foci within global health? For example, this manuscript presents a qualitative study conducted in a global health context. To what extent might the recommendations made apply for a study like this?

The language in the conclusion needs to be toned down a little. For example, '...do not always facilitate true understanding....'

	should be revised to ‘...may not always...’ (as your study has identified potential issues, but has not assessed whether these issues actually occurred in practice).
--	---

VERSION 1 – AUTHOR RESPONSE

Reviewer: 1

Dr. Justin Pulford, Liverpool School of Tropical Medicine

Comments to the Author:

1. Title

The title is too broad as stated as it implies an extensive evaluation of the barriers to informed consent, when the focus is specifically on the issue of translating informed consent documents (primarily) within the context of maternal health research trials.

The title has been updated as follows (Page 1, Lines 1-3):

“Understanding the language barriers to translating informed consent documents for maternal health trials in Zambia: a qualitative study”

2. Abstract

Based on points made below in response to the substantive text, some additional context may need to be included in the abstract (e.g. the explicit focus on trials) and some statements toned down. The authors should also include the number of participants and data analysis approaches in the ‘methods’ sub-section.

The abstract has been updated as follows (Page 2-3, Lines 17-50):

Abstract

Objective

Providing comprehensible information is essential to the process of valid informed consent. Recruitment materials designed by sponsoring institutions in English-speaking, high-income countries are commonly translated for use in global health studies in other countries; however, key concepts are often missed, misunderstood or “lost in translation”. The aim of this study was to explore the language barriers to informed consent, focusing on the challenges of translating recruitment materials for maternal health studies into Zambian languages.

Design

We used a qualitative approach, which incorporated a multi-stakeholder workshop (11 participants), in-depth interviews with researchers and translators (8 participants), and two community-based focus groups with volunteers from community advisory boards (20 participants). Content analysis was used to identify terms commonly occurring in recruitment materials prior to the workshop. The framework analysis approach was used to analyse interview data, and a simple inductive thematic analysis approach used to analysis focus group data.

Setting

The study was based in Lusaka, Zambia.

Results

The workshop highlighted difficulties in translating research terms and pregnancy-specific terms, as well as widespread concern that current templates are too long, use overly formal language, and are designed with little input from local teams. Framework analysis of in-depth interviews identified barriers to participant understanding relating to design and development of recruitment materials, language, local context, and communication styles. Focus group participants confirmed these findings and suggested potential solutions to ensure the language and content of recruitment materials can be

better understood.

Conclusion

Our findings demonstrate that the way in which recruitment materials are currently designed, translated, and disseminated, may not enable potential trial participants to fully understand the information provided. Instead of using overly complex institutional templates, recruitment materials should be created through an iterative and interactive process that provides truly comprehensible information in a format appropriate for its intended participants.

3. Introduction

It is still not entirely clear whether the authors are focusing on translation of informed consent materials in a trial context or in global health research more broadly. Much of the argument is strongest within a trial context, e.g. the use of 'medico-legal language' is perhaps most pronounced in medical trials as compared to other study designs/research foci, and the authors specifically refer to 'trial sponsors' and 'trial participants' in places. However, there is also reference to a 'maternal health' focus (which includes, but is not limited to, research trials). If the work primarily relates to medical trials conducted within a maternal health context, then this needs to be made explicit from the outset. I would suggest making this statement within the sentence beginning 'This study evaluates...' (line 59) and possibly within the manuscript title. Having made this focus explicit, the authors can then frame their argument within this context and, where appropriate, note issues and findings that may be relevant to informed consent processes more broadly (rather than imply stated issues are relevant in all cases of informed consent irrespective of research design/study focus). This would prevent/reduce some of the over generalisation that is still present when the explicit context is not made clear (i.e. informed consent language and protocols are often more strict within a medical trial context and issues associated with translating complex medical terminology are often more apparent in these cases). Other examples of over generalisation (not related to study design) are also apparent. For example, not all consent forms need to be translated (depends on country, type of participants etc). The manuscript has been updated as follows:

Page 5, Lines 70-73

"This study evaluates an example within the context of a maternal health trial conducted in Lusaka, Zambia, which specifically explores how language barriers, and issues surrounding translation of recruitment materials, may impact upon informed consent"

Page 5, Lines 79-83

"The participant information leaflets and consent forms (recruitment materials) used for enrolling participants into clinical trials conducted in low or middle income countries are often designed by sponsoring institutions based in high-income countries⁸ and are therefore more likely to meet the needs of trial sponsors and ethical review boards rather than those of the intended participants."

4. Methods

The focus of the paper is specifically on translating informed consent documents within a Zambian context; however, in the 'sampling strategy' section it is stated that 'recruitment materials' were obtained from researchers working in Zambia and 'neighbouring countries'. Why were documents from neighbouring countries included and would the workshop participants have been able to critique these, especially any related back translations?

The documents were in English and formed part of the sample of recruitment materials analysed by the study lead to identify the most commonly occurring terms, as a basis for discussion during the workshop. The manuscript has been updated as follows (Page 9, lines 114-142):

"The cross-sectional sample of recruitment materials used during Phase One was obtained by inviting researchers working in Zambia to submit English language examples of recruitment materials they had previously developed (and subsequently translated) to inform individuals considering participation in their research studies (predominantly clinical trials). Researchers were identified via ongoing research being conducted at University Teaching Hospital, Lusaka, ongoing research conducted by the Department of Women and Children's Health at King's College London, and via the Global

Women's Research Society international conference. Researchers were asked to provide English language versions of participant information leaflets which were collated, read, and analysed by the study lead (ABG). All relevant examples of recruitment materials provided were included in the total sample of 13 documents. Summative content analysis (see Data analysis section) was used to identify the most commonly occurring terms related to research and pregnancy (details shown in Supplementary Table 1). These terms were organised into relevant themes such as pregnancy-specific terms, research concepts and confidentiality (Supplementary Table 1). The workshop focussed on how these different English terms could be translated for a Zambian population, and the potential difficulties that might be encountered when doing so. Through our discussion with workshop participants, we were able to identify which commonly occurring terms were most difficult to translate. Contemporaneous group notes were made on flip charts during this process. In addition, the independently performed back translations of participant information leaflets (translated as part of the timing of delivery in pre-eclampsia trial conducted in Zambia) were also discussed. These leaflets had been translated from English, into Nyanja, and then back to English. The discussion focussed on comparing and contrasting the original English versions with the back-translated versions."

5. The data collection methods/procedures for the stage one workshop are not well described in the methodology. Rather, the description extends only to the sampling strategy with details re workshop process coming later in the 'results' section (e.g. this is where we are first introduced to the review of back translated documents).

The manuscript has been updated as follows (Page 9, lines 114-142):

"The cross-sectional sample of recruitment materials used during Phase One was obtained by inviting researchers working in Zambia to submit English language examples of recruitment materials they had previously developed (and subsequently translated) to inform individuals considering participation in their research studies (predominantly clinical trials). Researchers were identified via ongoing research being conducted at University Teaching Hospital, Lusaka, ongoing research conducted by the Department of Women and Children's Health at King's College London, and via the Global Women's Research Society international conference. Researchers were asked to provide English language versions of participant information leaflets which were collated, read, and analysed by the study lead (ABG). All relevant examples of recruitment materials provided were included in the total sample of 13 documents. Summative content analysis (see Data analysis section) was used to identify the most commonly occurring terms related to research and pregnancy (details shown in Supplementary Table 1). These terms were organised into relevant themes such as pregnancy-specific terms, research concepts and confidentiality (Supplementary Table 1). The workshop focussed on how these different English terms could be translated for a Zambian population, and the potential difficulties that might be encountered when doing so. Through our discussion with workshop participants, we were able to identify which commonly occurring terms were most difficult to translate. Contemporaneous group notes were made on flip charts during this process. In addition, the independently performed back translations of participant information leaflets (translated as part of the timing of delivery in pre-eclampsia trial conducted in Zambia) were also discussed. These leaflets had been translated from English, into Nyanja, and then back to English. The discussion focussed on comparing and contrasting the original English versions with the back-translated versions."

6. In the 'data analysis' section the authors state that 'no formal analysis of workshop data was performed', yet (in addition to findings from the content analysis) a summary of key points from the workshop discussion and examples of flawed back translations emanating from the workshop are presented in the results. As this is presented as data then how it was derived needs to be made clearer in the methods (see previous point). I.e. what you 'did' in the workshop needs to be described in more detail in the methods and text pertaining to the workshop in the results section should be limited to what you 'found'.

The manuscript has been updated as follows (Page 9, lines 114-142):

"The cross-sectional sample of recruitment materials used during Phase One was obtained by inviting researchers working in Zambia to submit English language examples of recruitment materials they

had previously developed (and subsequently translated) to inform individuals considering participation in their research studies (predominantly clinical trials). Researchers were identified via ongoing research being conducted at University Teaching Hospital, Lusaka, ongoing research conducted by the Department of Women and Children's Health at King's College London, and via the Global Women's Research Society international conference. Researchers were asked to provide English language versions of participant information leaflets which were collated, read, and analysed by the study lead (ABG). All relevant examples of recruitment materials provided were included in the total sample of 13 documents. Summative content analysis (see Data analysis section) was used to identify the most commonly occurring terms related to research and pregnancy (details shown in Supplementary Table 1). These terms were organised into relevant themes such as pregnancy-specific terms, research concepts and confidentiality (Supplementary Table 1). The workshop focussed on how these different English terms could be translated for a Zambian population, and the potential difficulties that might be encountered when doing so. Through our discussion with workshop participants, we were able to identify which commonly occurring terms were most difficult to translate, Contemporaneous group notes were made on flip charts during this process. In addition, the independently performed back translations of participant information leaflets (translated as part of the timing of delivery in pre-eclampsia trial conducted in Zambia) were also discussed. These leaflets had been translated from English, into Nyanja, and then back to English. The discussion focussed on comparing and contrasting the original English versions with the back-translated versions."

Page 10, lines 179-186:

Content analysis was used to analyse recruitment materials as a recognised method of rapidly identifying commonly occurring language. A summative approach was taken, identifying the number of times that pregnancy related phrases and research terms arose in the sample of recruitment materials.²¹ During the Phase One workshop (conducted in English), participants discussed the interpretation of these commonly occurring English terms, and explored how they might be translated into Nyanja and Bemba. Key outputs from the discussion notes were used to inform the design of Phases Two and Three.

7. Results

The content under 'phase one: initial workshop' should be amended based on comments above (i.e. content describing process moved to the methods section). Content in remaining sections OK as is. The manuscript has been updated accordingly, with the relevant section moved to methods section (Page 9, lines 114-142):

"The cross-sectional sample of recruitment materials used during Phase One was obtained by inviting researchers working in Zambia to submit English language examples of recruitment materials they had previously developed (and subsequently translated) to inform individuals considering participation in their research studies (predominantly clinical trials). Researchers were identified via ongoing research being conducted at University Teaching Hospital, Lusaka, ongoing research conducted by the Department of Women and Children's Health at King's College London, and via the Global Women's Research Society international conference. Researchers were asked to provide English language versions of participant information leaflets which were collated, read, and analysed by the study lead (ABG). All relevant examples of recruitment materials provided were included in the total sample of 13 documents. Summative content analysis (see Data analysis section) was used to identify the most commonly occurring terms related to research and pregnancy (details shown in Supplementary Table 1). These terms were organised into relevant themes such as pregnancy-specific terms, research concepts and confidentiality (Supplementary Table 1). The workshop focussed on how these different English terms could be translated for a Zambian population, and the potential difficulties that might be encountered when doing so. Through our discussion with workshop participants, we were able to identify which commonly occurring terms were most difficult to translate, Contemporaneous group notes were made on flip charts during this process. In addition, the independently performed back translations of participant information leaflets (translated as part of the timing of delivery in pre-eclampsia trial conducted in Zambia) were also discussed. These leaflets had

been translated from English, into Nyanja, and then back to English. The discussion focussed on comparing and contrasting the original English versions with the back-translated versions.”

8. Discussion

A lot of the content under ‘strengths and weaknesses’ is related to the issue of positionality in a qualitative research context. Positionality statements would typically be found in a the ‘methods’ section. Thus, the authors could consider shifting some of the material from here to a specific ‘positionality’ sub-section in the methods.

I would be happy to move this to the methods section at the editor’s discretion

The issue of context could also be discussed in this section. I.e. if this study is primarily framed within the context of informed consent carried out in maternal health trials, then to what extent might the findings relate to informed consent in other study designs/study foci within global health? For example, this manuscript presents a qualitative study conducted in a global health context. To what extent might the recommendations made apply for a study like this?

The manuscript has been updated as follows (Page 21, lines 415-421):

“In this study, we focus on the example of translating recruitment materials for a maternal health-related clinical trial in Zambia. It is possible, that the specific challenges described by participants in this study, may not necessarily to apply to other study designs or contexts, thereby limiting the generalisability of our findings. However, by sharing the key learning points identified from this qualitative study, it may prompt any individual involved in translating or using participant information in similar settings to critically review the language they use, and whether it is appropriate and comprehensible to its intended audience.”

The language in the conclusion needs to be toned down a little. For example, ‘...do not always facilitate true understanding...’ should be revised to ‘...may not always...’ (as your study has identified potential issues, but has not assessed whether these issues actually occurred in practice).

The manuscript has been updated as follows (Page 21-22, lines 423-429):

“Our study has identified that current methods of designing and translating recruitment materials for potential research participants in maternal health studies in Zambia, may not always facilitate true understanding, and therefore may not serve the needs of their intended recipients.”

VERSION 2 – REVIEW

REVIEWER	Pulford, Justin Liverpool School of Tropical Medicine, Department of International Public Health
REVIEW RETURNED	11-Jan-2024
GENERAL COMMENTS	The suggested revisions have been addressed very well.